# Omitting Hyperventilation in Electroencephalogram during the COVID-19 Pandemic May Reduce Interictal Epileptiform Discharges in Patients with Juvenile Myoclonic Epilepsy

**DOI:** 10.3390/brainsci12060769

**Published:** 2022-06-11

**Authors:** Keisuke Hatano, Ayataka Fujimoto, Keishiro Sato, Takamichi Yamamoto, Hideo Enoki

**Affiliations:** 1Comprehensive Epilepsy Center, Seirei Hamamatsu General Hospital, 2-12-12 Sumiyoshi, Nakaku, Hamamatsu 430-8558, Shizuoka, Japan; hatakenosuke@gmail.com (K.H.); k-sato@sis.seirei.or.jp (K.S.); taka-yamamd@sis.seirei.or.jp (T.Y.); enokih.neuropediatr@gmail.com (H.E.); 2Department of Neurosurgery, Seirei Hamamatsu General Hospital, 2-12-12 Sumiyoshi, Nakaku, Hamamatsu 430-8558, Shizuoka, Japan

**Keywords:** electroencephalogram (EEG), hyperventilation (HV), interictal epileptiform discharge (IED), juvenile myoclonic epilepsy (JME), coronavirus disease 2019 (COVID-19)

## Abstract

Background: To prevent the spread of coronavirus disease 2019 (COVID-19), hyperventilation (HV) activation has been avoided in electroencephalograms (EEGs) since April 2020. The influence of omitting HV in EEG on epilepsy diagnosis remains uncertain for patients with epilepsies other than child absence epilepsy. We hypothesized that EEGs with HV would show more interictal epileptiform discharges (IEDs) than EEGs without HV in patients with juvenile myoclonic epilepsy (JME). Methods: We reviewed the EEGs of seizure-free patients with JME who underwent EEG, both with and without HV, from January 2019 to October 2021, in our institution, and compared IEDs between EEG with and without HV. Results: This study analyzed 23 JME patients. The IED-positive rate was significantly higher in EEG with HV (65.2%) than in EEG without HV (34.8%, *p* = 0.016). The mean ± standard deviation number of IEDs per minute was significantly larger during HV (1.61 ± 2.25 × 10^−1^) than during non-activation of both first EEG (0.57 ± 0.93 × 10^−1^, *p* = 0.039) and second EEG (0.39 ± 0.76 × 10^−1^, *p* = 0.009). Conclusions: In JME patients, performing HV during EEG may increase IEDs and appears to facilitate the accurate diagnosis of epilepsy.

## 1. Introduction

Due to the coronavirus disease 2019 (COVID-19) pandemic, the use of electroencephalograms (EEGs) has been reduced, since these are usually performed in a closed room with insufficient ventilation [1,2]. In addition, measures, such as wearing masks, for both patients and technologists and disinfection of equipment before taking the EEG have been recommended by the American Academy of Clinical Neurophysiology (ACNS) and the International League Against Epilepsy (ILAE) [1]. Moreover, since hyperventilation (HV) carries a risk of triggering coughing and requires the patient to remove any mask to reduce the partial pressure of arterial carbon dioxide (PaCO_2_), some authors have recommended use of HV only for patients in whom absence epilepsies are suspected [3,4]. Our institution has been conducting HV-free EEG examinations in principle since April 2020.

HV has been reported to induce interictal epileptiform discharges (IEDs) in ~80% of cases of idiopathic generalized epilepsy (IGE) and ~50% of cases of symptomatic generalized epilepsy [5,6,7,8] and was routinely performed before the COVID-19 pandemic. On the other hand, several reports have suggested that an increase in or appearance of IEDs during HV was less than 10% when most cases were focal epilepsy, representing the basis for omitting HV during the COVID-19 pandemic [4,9].

However, no reports have examined the impact of omitting HV in EEG during the COVID-19 pandemic, particularly in terms of the risk of overlooking an epilepsy diagnosis. Is HV-free EEG really no problem, even for patients with any IGE other than child absence epilepsy? As a representative IGE, we focused on juvenile myoclonic epilepsy (JME), which accounts for 18 to 37.3% of IGE [10,11]. We hypothesized that EEGs with HV would show more IEDs than EEGs without HV in JME patients. To test this hypothesis, we retrospectively reviewed data from JME patients who underwent EEGs before and after the COVID-19 pandemic and compared the findings between EEGs with and without HV. Our institution showed almost no loss of JME patients, because we followed-up epilepsy patients using an “Epi passport”, a unique booklet for information sharing among regional epilepsy networks [12].

## 2. Materials and Methods

### 2.1. Participants and Study Design

The ethics committee at Seirei Hamamatsu General Hospital approved this study protocol and waived the requirement for written informed consent because this study was considered to pose minimal risk to participants. This single-site, retrospective, observational study recruited JME patients who had undergone EEGs both with and without HV between January 2019 and October 2021 at the Comprehensive Epilepsy Center, Seirei Hamamatsu General Hospital. Based on the report by Pedersen et al. [13] and the 2017 classification of the ILAE [14,15,16], criteria for JME were as follows: (1) clinical evidence of myoclonic jerks with or without typical absence seizures or generalized tonic-clonic seizures (GTCS) and (2) normal intelligence and neurological findings on neurological examination. Exclusion criteria were: (1) under 16 years old; (2) epilepsy associated with brain hypoxia, metabolic, or progressive diseases; (3) before starting anti-seizure medication (ASM) or during up-dosing; (4) EEG measurements not performed using the international 10–20 system; or (5) a request from the patient to opt out of participation in the study.

As for demographic and clinical data, we obtained sex, age (at onset of epilepsy and at first/second EEG), number and types of ASMs, seizure types (typical absence seizures or GTCS), seizure-free status for more than 2 years, drowsiness or sleep during EEG, and time measured by EEG.

### 2.2. EEG Measurement

All EEGs were obtained using Neurofax EEG 1200 systems (EEG-1218 (serial no. 49) and EEG-1200 (serial no. 526); Nihon Kohden, Tokyo, Japan). Before the COVID-19 pandemic, two opening and closing eye tests, photic stimulation (PS) at 6, 8, 12, 15, 18, and 20 Hz, and 3 min under HV load were routinely performed in sequence. In HV, the patient was required to reach a respiration rate of 20 to 30 breaths/min. After HV, EEG was continued for at least 5 min and as long as possible until the patient fell asleep. Since April 2020, EEG has been performed without HV and wearing a mask has been mandatory (Figure 1). If a patient underwent EEG more than two times during the study period, we analyzed the two EEGs closest to April 2020.

### 2.3. Outome Measurement

Abnormal EEG configurations included as IEDs in this study were: spike and slow wave (SW); poly-spike and slow wave; and spike burst (Figure 2). Sharp waves (duration, 70 to 200 milliseconds) were also included along with spike waves (duration, 20 to 70 milliseconds). SWs were subdivided into three groups according to the distribution (Figure 2): (1) generalized SW (GSW); (2) bilaterally symmetrical but not generalized SW (bilateral SW); or (3) lateralized SW (focal). Among the SWs, 6 Hz SWs were excluded because these are considered a pattern of uncertain significance [17]. Since diffuse slow waves induced by HV are considered to be a physiologic pattern [17], slow waves alone were also excluded from IEDs.

As primary outcome measures, we investigated the presence or absence of IED and conducted comparisons between EEG with and without HV. As secondary outcome measures, the presence or absence of IEDs during activations (PS, HV, drowsiness, and sleep) and number of IEDs during an EEG examination were assessed. We expected that the time of HV-free EEGs would be shorter in duration than that of EEGs with HV because the time required to perform HV would be omitted. This may lead to overestimation of the number of IEDs in EEGs with HV. Therefore, we also investigated the number of IEDs per minute. “Number of IEDs/min” means the value obtained by dividing the number of IEDs during an EEG examination by the duration of EEG.

We also reviewed IED morphology (GSW, bilateral SW, focal, and spike burst) and compared EEG with and without HV. Moreover, to directly assess increases in IEDs by HV, the number of IEDs per minute during HV procedure was compared to that during non-activation (other than PS, HV, drowsiness, and sleep) in the first and second EEGs. “Number of IEDs/min limited to HV” means the value obtained by dividing the number of IEDs during HV procedure by the duration of HV.

### 2.4. Statistical Analysis

For intergroup comparisons of categorical data, we used the McNemar’s test when the sample size was small. To compare continuous data, the paired *t*-test was used. Two-sided *p*-values < 0.05 were considered statistically significant. All statistical analyses were performed using Stata/SE Version 14.0 (StataCorp LP, College Station, TX, USA).

## 3. Results

Forty-two JME patients over 16 years old underwent EEG twice or more during the target period. Ten patients were measured using methods other than the international 10–20 system, one patient was measured by long-term video EEG, one patient performed HV in both EEGs, four patients did not perform HV in either of the two EEGs, and three patients underwent the first EEG before starting ASM or during up-dosing. These 19 patients were, thus, excluded and the remaining 23 participants were analyzed in the current study.

### 3.1. Patient Characteristics

Mean age at seizure onset was 16.1 years (standard deviation (SD) = 9.9 years) and 17 patients (73.9%) were female. GTCS was found in addition to myoclonic jerk in all cases, but none showed typical absence seizures. Age at EEG test, percentage of patients with freedom from seizures, and presence or absence of drowsiness and sleep during EEG were similar between EEGs with and without HV (Table 1). Mean time in EEG was significantly longer for EEGs with HV (21.4 ± 2.9 min) than for EEGs without HV (16.7 ± 3.0 min, *p* < 0.0001). Mean duration between the two EEGs was 15.0 ± 3.7 months.

### 3.2. Outcome Measures

The presence of IEDs was found in 15 patients (65.2%) for EEG with HV and 8 patients (34.8%) for EEG without HV, showing a significant difference between the two EEG tests (*p* = 0.016). During HV, nine patients (39.1%) experienced IEDs. The presence rates of IED during PS, drowsiness, or sleep were similar between the two EEG tests (Table 2).

The mean number of IEDs per EEG was 1.61 ± 1.77 for EEGs with HV and 0.91 ± 1.53 for EEGs without HV (*p* = 0.073). The mean number of IEDs/min did not differ significantly between EEGs with and without HV (Table 2). No clinical seizures were seen in any EEGs.

Divided by the morphology of IEDs, the mean number of bilateral SW per EEG differed significantly between EEGs with HV (0.96 ± 1.43) and without HV (0.48 ± 0.99, *p* = 0.046). Mean numbers of GSW, focal SW, and spike burst per EEG were similar between both EEG tests (Figure 3). The proportion of IEDs that continued for more than 2 s was 15.8% (6/38 IEDs) for EEGs with HV and 6.3% (1/16 IEDs) for EEGs without HV.

Mean number of IEDs/min limited to HV was 1.61 ± 2.25 × 10^−1^. Mean number of IEDs/min limited to non-activation time was 0.57 ± 0.93 × 10^−1^ for the first EEG with HV and 0.39 ± 0.76 × 10^−1^ for the second EEG without HV (Figure 4). The mean number of IEDs/min when limited to HV was significantly larger than that limited to non-activation, at both first and second EEGs (*p* = 0.039 and 0.009, respectively).

## 4. Discussion

The present study revealed that the rate of IEDs in JME patients was significantly higher in EEGs with HV (65.2%) than in EEGs without HV (34.8%). Moreover, the number of IEDs/min when limited to HV was also shown to be significantly higher than that limited to non-activation. These results support our hypothesis that EEGs with HV would have more IEDs than EEGs without HV in JME patients.

The incidence rate of IEDs in EEGs for JME patients has been reported as 70 to 93.7% [18,19,20]. In the present study, the rate of IEDs was slightly lower (65.2%), even for EEGs with HV. This result may be associated with the fact that this study only included patients with good seizure control and excluded slow waves alone and 6 Hz SWs. This may be for the same reason that almost all IEDs in this study did not continue more than 2 s. Arntsen et al. [21] suggested that IEDs lasting more than 3 s are related to poor seizure control in JME patients. Since fragments of GSW were defined as brief (<2 s) IEDs without clinical signs, which may not necessarily be generalizable [22,23], 84.2% of GSWs in EEGs with HV and 93.7% without HV in this study were considered as fragments of GSW.

The incidence rate of IEDs by HV reportedly varies from 6.6 to 100% [4,8,18,20], although reports in which the majority of cases had focal epilepsy tended to show a low rate of HV-induced IEDs < 10% [4,8]. The current result that 39.1% of participants had IEDs during HV seems consistent with these previous reports, considering that this study was limited to JME patients.

Performing HV for patients with suspected JME may help to diagnose epilepsy accurately, because this study showed IEDs increased with the addition of HV in EEG. Therefore, if a patient suspected of JME does not have any IEDs in a HV-free EEG, performing HV may be a reasonable choice in the next EEG. Although the mechanisms of IED activation by HV have not been elucidated, the main cause is considered to be a decrease in PaCO_2_ induced by HV. This decline in PaCO_2_ reduces cerebral blood flow by vasoconstriction of the cerebral arteries and appears to activate IEDs [2]. However, the same mechanism may not occur when a patient wears a face mask during HV tests. Conversely, HV with a face mask has been documented to dilate cerebral arteries by increasing PaCO_2_ [2] or induce hypoxia [24]. Evidence about the influence of HV with face masks on increasing IEDs remains limited, although HV with face masks has been reported to increase excitability in the brain network [25]. Therefore, even during the COVID-19 outbreak, patients might need to remove their mask during HV. As a matter of course, given the risk of spreading COVID-19, whether to perform HV for patients with suspected JME and whether to remove the patient’s face mask, should be given careful consideration. Prior to HV, we should confirm that the patient has no symptoms related to COVID-19, such as fever, coughing, or malaise, and no close contact with any COVID-19-positive individuals. Depending on the situation, performing HV after confirming a negative polymerase chain reaction (PCR) for SARS-CoV-2 may be considered.

Several limitations were present in the current study. Our study was retrospective and had a small sample size of only 23 JME patients. In this study, the number of IEDs per EEG was similar between EEG with and without HV, but larger sample sizes may have made a difference. In addition, patients in this study were limited to seizure-free patients from a single institution. The results, thus, may not be generalizable to all JME patients. Further research is warranted to increase the external validity of these results.

## 5. Conclusions

In JME patients, performing HV during EEG may increase IEDs and facilitate the accurate diagnosis of epilepsy. Even during the COVID-19 pandemic, performing HV may be considered for patients with suspected JME. Given the current results, studies of patients with new-onset epilepsy or large population-based studies across multiple institutions appear warranted.

## Figures and Tables

**Figure 1 brainsci-12-00769-f001:**
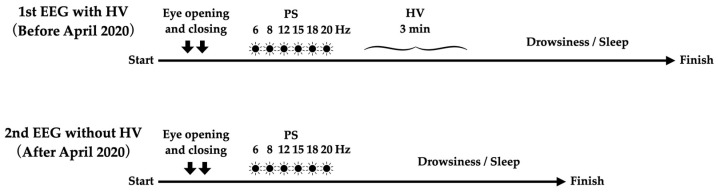
Flow of EEG examination. In our institution, HV was routinely performed before the COVID-19 outbreak, but was omitted after April 2020.

**Figure 2 brainsci-12-00769-f002:**
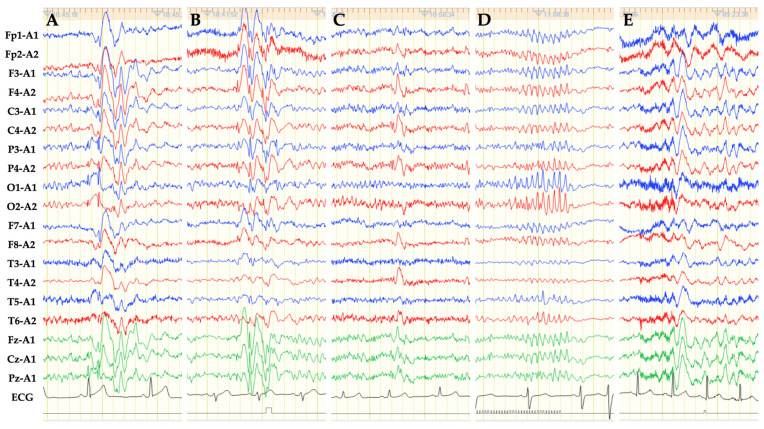
EEG of referential derivation using the 10–20 international system shows various forms of IEDs. (**A**) GSWs are characterized by bilateral, synchronous SWs found in all regions. (**B**) Bilateral SWs are defined as bilateral, synchronous SW not found in all regions. This representative EEG shows bilateral SWs in the areas except the temporal region. (**C**) Lateralized SWs (focal) show SW only on the left or right side. (**D**) Spike or sharp burst refers to consecutive spike or sharp waves. (**E**) Slow waves without preceding spike or sharp waves were excluded from IEDs in this study. Sensitivity = 10 μV/mm; time constant = 0.3 s; high frequency filter = 60 Hz; paper speed = 3 cm/s.

**Figure 3 brainsci-12-00769-f003:**
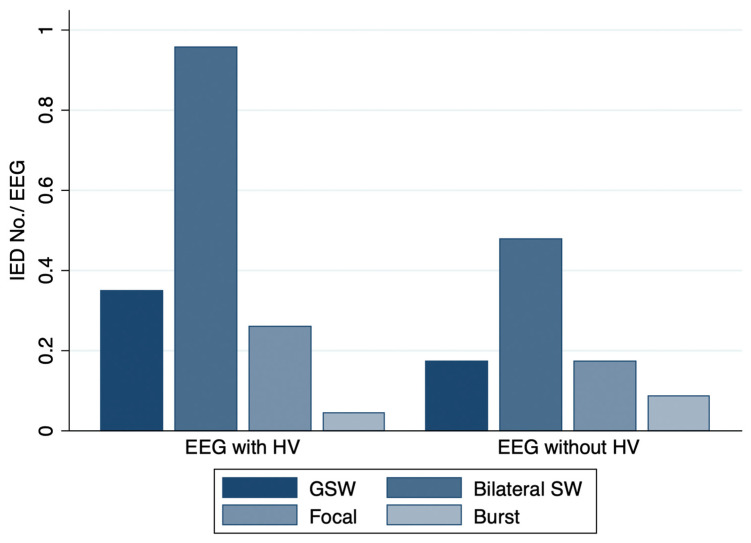
Histogram illustrating the number of IEDs subdivided by IED morphology. Mean ± SD number of GSW, bilateral SW, focal SW, and spike/sharp burst per EEG were 0.35 ± 0.78, 0.96 ± 1.43, 0.26 ± 0.62, and 0.09 ± 0.29 for EEG with HV and 0.17 ± 0.65, 0.48 ± 0.99, 0.17 ± 0.65, and 0.04 ± 0.21 for HV-free EEG, respectively.

**Figure 4 brainsci-12-00769-f004:**
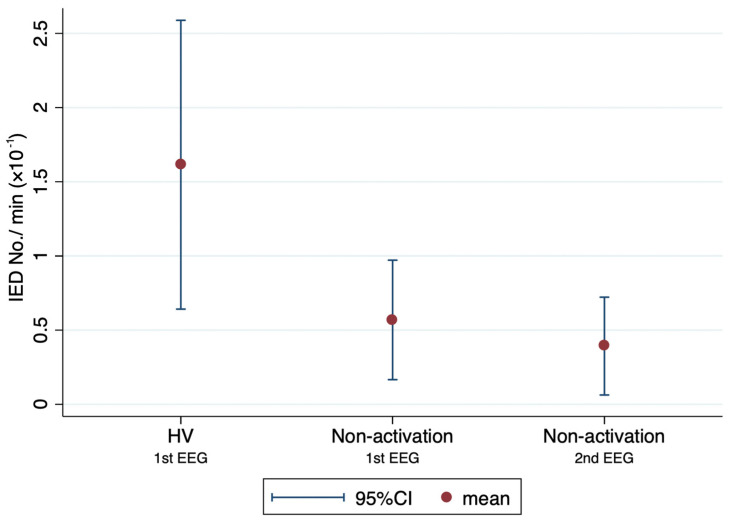
Mean number of IEDs/min limited to HV or non-activation. Dots and bars represent mean and 95% confidence interval (CI), respectively. Mean number of IEDs/min limited to HV was 1.61 × 10^−1^ (95%CI, 0.64–2.59 × 10^−1^) and that limited to non-activation was 0.57 × 10^−1^ (95%CI, 0.17–0.97 × 10^−1^) at first EEGs and 0.39 × 10^−1^ (95%CI, 0.06–0.72 × 10^−1^) at second EEGs. Number of IEDs/min limited to HV was significantly higher than that limited to non-activation at both first and second EEGs (*p* = 0.039 and 0.009, respectively).

**Table 1 brainsci-12-00769-t001:** Patient and EEG characteristics at the two EEG measurements.

	EEG with HV (*n* = 23)	EEG without HV (*n* = 23)	*p* Value
Age at EEG, years *	29.8 ± 13.6	31.1 ± 13.5	0.75
Seizure-free > 2 years, %	91.3%	91.3%	1
Time in EEG, min *	21.4 ± 2.9	16.7 ± 3.0	<0.0001
Drowsiness, %	87.0%	91.3%	1
Sleep, %	60.9%	60.9%	1

* Expressed as mean ± standard deviation.

**Table 2 brainsci-12-00769-t002:** Outcome measures comparing EEGs with HV and without HV.

	EEG with HV	EEG without HV	*p* Value
IED-yes, % (*n*)	65.2% (15/23)	34.8% (8/23)	0.016
during PS, % (*n*)	21.7% (5/23)	17.4% (4/23)	1
during HV, % (*n*)	39.1% (9/23)	-	
during drowsiness, % (*n*)	10.0% (2/20)	9.5% (2/21)	1
during sleep, % (*n*)	6.7% (1/15)	0% (0/14)	1
Number of IEDs/EEG *	1.6 ± 1.8	0.9 ± 1.5	0.073
Number of IED/min × 10^−1^ *	7.9 ± 9.2	5.5 ± 9.5	0.27

* Expressed as mean ± standard deviation.

## Data Availability

Not applicable.

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
