# Peer review of "Omitting Hyperventilation in Electroencephalogram during the COVID-19 Pandemic May Reduce Interictal Epileptiform Discharges in Patients with Juvenile Myoclonic Epilepsy"

_brainsci, 2022, doi:10.3390/brainsci12060769_

Round 1
Reviewer 1 Report
Major Issue -
For the analysis of the data, authors should use methods for paired data - since the HV and non-HV measures are taken from the same subject. It would be appropriate to use McNemar's test and paired t-tests.
Other Notes/Questions:
1. JME was chosen as representative IGE - maybe add a sentence as to why it is representative
2. In the first paragraph of the Results, it enumerates those who was excluded - it says 19 patients were excluded, but it reads as if only 16 were excluded. My assumption is that 3 were excluded for first EEG before starting ASM and another 3 were excluded for first EEG before up-dosing. Could you please clarify this in the manuscript?
3. What is the distribution of time between EEGs for the patients? Could this be a possible confounder?
In the discussion you mention that EEGs with HV were done before EEGs without HV - I think this is good so you don't have to account for which was done first
Author Response
Major Issue -
For the analysis of the data, authors should use methods for paired data - since the HV and non-HV measures are taken from the same subject. It would be appropriate to use McNemar's test and paired t-tests.
→Thank you for pointing this out. We have revised the results of analysis by using McNemar’s test instead of χ2 test or Fisher’s exact test.
Other Notes/Questions:
- JME was chosen as representative IGE - maybe add a sentence as to why it is representative
→Thank you for suggesting the addition. We have added the sentence “we focused on juvenile myoclonic epilepsy (JME) which accounts for 18-37.3 % of IGE” in the Introduction.
- In the first paragraph of the Results, it enumerates those who was excluded - it says 19 patients were excluded, but it reads as if only 16 were excluded. My assumption is that 3 were excluded for first EEG before starting ASM and another 3 were excluded for first EEG before up-dosing. Could you please clarify this in the manuscript?
→Thank you for pointing out this discrepancy. We apology for making a mistake. We have correctly changed “1 patient did not perform HV in either of the two EEGs” to “4 patient did not perform HV in either of the two EEGs”.
- What is the distribution of time between EEGs for the patients? Could this be a possible confounder?
→Thank you for pointing out the important content. We have added the sentence “The mean duration between the two EEGs was 15.0±3.7 months.”. But we don’t think the duration between EEGs is a possible confounder because seizures were under control in all participants.
- In the discussion you mention that EEGs with HV were done before EEGs without HV - I think this is good so you don't have to account for which was done first
→We agree with this and have deleted the sentence “historical bias was present because all participants in this study underwent EEG with HV first, followed by EEG without HV. However, this historical bias seems negligible, as the present study included only patients with good control of seizures with constant ASMs at both time points of the two EEGs.” in Discussion.
Reviewer 2 Report
This is a well designed study with a clear question: Whether omitting hyperventilation during EEG affects the detection of IEDs in patients with JME. This is significant, given the police changes in performing routine EEG recordings during the COVID19 pandemic.
Hyperventilation is a well known and well studied activation method, however there is a large variability in the studies that tried to measure the exact effect of HV. This study, although based on a small number of patients, adds another result to the usefulness of HV in JME.
The results are clear and well presented. The limitations are identified and discussed by the authors.
The manuscript adds in the discussion of policies regarding HV during the pandemic.
Author Response
Thank you for your kind evaluation.
Reviewer 3 Report
1. What is the meaning of Sleep stage 1 and Sleep stage 2 in your manuscript?
2. Avoiding confusion meaning, "2s", "3s" should revise to "2 second" , "3 second". Please apply for another too.
3. Repeating words and sentences should not exist in the manuscript. There are given abbreviations many times for the same word, in the caption of Table 1,2, Figure1, 2.
For example, this abbreviation was given "hyperventilation (HV)" in Introduction Section. So when you talk about "hyperventilation" second time, just use "HV"
4. The author should provide pictures during performing experiments ( experiment of various conditions, EEG with HV, EEG without HV).
5. There are analyses of EEG signals in the study, but the authors don't give pictures of the analysis of EEG signals.
Giving the analysis of EEG with HV signal and EEG without HV signal is essential for the reader to make sense of what the author is trying to explain.
6. Please rewrite again and clarify their meaning, 20-30, 6-Hz. Do you mean 20 to 30, 6Hz?
7. First sentence of 2.2 Section, the more specific serial number of equipment is better to mention instead of the name of the company(Nihon Kohden, Tokyo, Japan)
8. What is PS refers to? Please give the whole meaning of the abbreviation "PS" first before using it.
9. Using special characters in the manuscript the reader might confuse about the meaning or not make sense. For example, in line 64, "...and/or generalized...".
10. Please double-check the grammar of the manuscript again, avoiding using long sentences with a lot of prepositions in the same sentence.
11. There are many styles of describing the same concept. For example, 15 patients, Nine patients. Please use only one style; writing numbers within a text is recommended.
12. 3.1 Section, what is the meaning of 73.9%? I recommend rewriting this sentence again "Mean age...........................................were female".
13. Please add more detail and description to Method Section and Result Section.
Moreover, the explanation of the method is hard to understand, so please write the used method with reason and avoid just reporting style in the method.
14. Discussion Section lines 172&173, incorrect citation. Please revise it to "...Arntsen et al.[19]...".
15. The conclusion is so short and does not have a complete meaning. Please add more and also include limitations and contributions at the end sentences in the Conclusions section.
Author Response
- What is the meaning of Sleep stage 1 and Sleep stage 2 in your manuscript?
→In Japan, sleep is conventionally classified into some stages. The sleep stage seen vertex sharp transient in EEG is sleep stage 1. The stage seen sleep spindle or K-complex is sleep stage 2. But this classification is not common worldwide and not important for activation of epileptiform discharges. To eliminate confusion, we removed “sleep stage” and only mentioned about the presence or absence of “sleep” in Table 1. Thank you for pointing out.
- Avoiding confusion meaning, "2s", "3s" should revise to "2 second" , "3 second". Please apply for another too.
→We agree with your suggestion. We have changed "2s" and "3s" to "2 seconds" and "3 seconds".
- Repeating words and sentences should not exist in the manuscript. There are given abbreviations many times for the same word, in the caption of Table 1,2, Figure1, 2.
For example, this abbreviation was given "hyperventilation (HV)" in Introduction Section. So when you talk about "hyperventilation" second time, just use "HV"
→As you pointed out, the explanation of the abbreviations was repeated too much. We have deleted abbreviations in the Table and Figure.
- The author should provide pictures during performing experiments ( experiment of various conditions, EEG with HV, EEG without HV).
→Thank you for your proposal. For clarity, we used a schema in new Figure 1 to explain the difference between EEG with HV and EEG without HV.
- There are analyses of EEG signals in the study, but the authors don't give pictures of the analysis of EEG signals.
Giving the analysis of EEG with HV signal and EEG without HV signal is essential for the reader to make sense of what the author is trying to explain.
→Thank you for your proposal. We have presented pictures of EEG classified by morphology of IEDs (GSW, bilateral SW, focal, spike burst, and slow waves) in new Figure 2. Since analysis of EEG with HV was the same way as that without HV, we don't think it is necessary to make a separate presentation depending on the presence or absence of HV.
- Please rewrite again and clarify their meaning, 20-30, 6-Hz. Do you mean 20 to 30, 6Hz?
→We have changed “20-30” to “20 to 30” and “6-Hz” to “6 Hz”. Thank you for teaching.
- First sentence of 2.2 Section, the more specific serial number of equipment is better to mention instead of the name of the company(Nihon Kohden, Tokyo, Japan)
→We agree with your suggestion. We have added the serial number such as “Neurofax EEG 1200 systems (EEG-1218 [serial no. 49] and EEG-1200 [serial no.526], Nihon Kohden, Tokyo, Japan)”
- What is PS refers to? Please give the whole meaning of the abbreviation "PS" first before using it.
→This sentence “photic stimulation (PS at 6, 8, 12, 15, 18, and 20 Hz)” in 2.2. Section may be difficult to understand. We have revised this sentence to “photic stimulation (PS) at 6, 8, 12, 15, 18, and 20 Hz”. Thank you for pointing this out.
- Using special characters in the manuscript the reader might confuse about the meaning or not make sense. For example, in line 64, "...and/or generalized...".
→We agree with your suggestion. We have changed "...and/or generalized..." to “… or generalized…”.
- Please double-check the grammar of the manuscript again, avoiding using long sentences with a lot of prepositions in the same sentence.
→Thank you for your suggestion. We asked FORTE (a professional language editing service) for proofreading again. FORTE revised partially our manuscript but not completely. FORTE may revise further, but we submitted our revision manuscript because deadline(within 7 days) has come. We have converted long sentences to several short sentences as bellow.
- “Since we expected the time of HV-free EEGs would be shorter than that of EEGs with HV, we also investigated the number of IEDs per minute” in 2.3 Section.
→“We expected that the time of HV-free EEGs would be shorter than that of EEGs with HV because the time to perform HV would be omitted. This can lead to overestimation of the number of IEDs in EEGs with HV. Therefore, we also investigated the number of IEDs per minute”.
- “This may be for the same reason that almost all IEDs in this study did not continue more than 2 seconds, because Arntsen et al. [21] discussed that IEDs lasting more than 3 seconds are related to poor seizure control in JME patients” in 2.3 Section.
→“This may be for the same reason that almost all IEDs in this study did not continue more than 2 seconds. Arntsen et al. [21]discussed that IEDs lasting more than 3 seconds are related to poor seizure control in JME patients” in Discussion.
- “Although the mechanisms of IED activation by HV have not been elucidated, the main hypothesis is that HV induces a decrease in PaCO2, which reduces cerebral blood flow by vasoconstriction of the cerebral arteries [2].” in Disucussion.
→“Although the mechanisms of IED activation by HV have not been elucidated, the main cause is considered to be a decrease in PaCO2 induced by HV. The decline of PaCO2 reduces cerebral blood flow by vasoconstriction of the cerebral arteries and appears to activate IEDs [2].”
- There are many styles of describing the same concept. For example, 15 patients, Nine patients. Please use only one style; writing numbers within a text is recommended.
→Thank you for your suggestion. We changed the sentence “Nine patients (39.1%) experienced IEDs during HV.” to “During HV, 9 patients (39.1%) experienced IEDs.” in 3.2. Section.
- 1 Section, what is the meaning of 73.9%? I recommend rewriting this sentence again "Mean age...........................................were female".
→We have revised the sentence ” Mean age … and 17 patients (73.9%) were female.” to ”Mean age … and 17 of 23 patients (73.9%) were female.” in the 3.1 Section.
- Please add more detail and description to Method Section and Result Section.
Moreover, the explanation of the method is hard to understand, so please write the used method with reason and avoid just reporting style in the method.
→Thank you for pointing out. We have revised as bellow.
- We have added explanation about the definition of IEDs in this study. For example, “Sharp wave (duration of 70 to 200 milliseconds) was also included as well as spike wave (duration of 20 to 70 milliseconds)” in 2.3 Section.
- The reasons of exclusion of slow wave alone and 6 Hz SW were added like “Among the SWs, 6 Hz SW was excluded because it is considered as pattern of uncertain significance [17]. Since diffuse slow wave induced by HV is considered as physiologic pattern [17], slow wave alone was also excluded from IEDs” in 2.3 Section.
- In 2.3. Section, we have added doing an evaluation of IED morphology, using the sentence “We also reviewed IED morphology (GSW, bilateral SW, focal, and spike burst) and compared EEG with and without HV.”
- We have matched the order written in methods and results (presence or absence of IED→IEDs during activation、the number of IEDs、number of IEDs/min→IED morphology→number of IEDs/min limited to HV).
- We think it may be difficult to distinguish between “number of IEDs/min” and “number of IEDs/min limited to HV”. Therefore, we have added the two sentences in the 2.3 Section as bellow, and changed “number of IEDs per minute” to ”number of IEDs/min” and “number of IEDs per minute limited to HV” to ”number of IEDs/min limited to HV” in 3.2 Section.
・“Number of IEDs/min” means the value obtained by dividing the number of IEDs during an EEG examination by the EEG time.
・“Number of IEDs/min limited to HV” means the value obtained by dividing the number of IEDs during HV procedure by the HV time.
- Discussion Section lines 172&173, incorrect citation. Please revise it to "...Arntsen et al.[19]...".
→We have changed the place of citation from “Arntsen et al. …in JME patients [21]” to “Arntsen et al. [21] discussed …”.
- The conclusion is so short and does not have a complete meaning. Please add more and also include limitations and contributions at the end sentences in the Conclusions section.
→We agree with your suggestion. We have added the sentences “Even during the COVID-19 pandemic, performing HV may be considered for patients with suspected JME. Given the current results, studies of patients with new onset epilepsy or large population-based studies across multiple institutions appear warranted.”